# The Role of Hypoxia-Inducible Factor 1 Alpha in Acute-on-Chronic Liver Failure

**DOI:** 10.3390/ijms25031542

**Published:** 2024-01-26

**Authors:** Marcus M. Mücke, Nihad El Bali, Katharina M. Schwarzkopf, Frank Erhard Uschner, Nico Kraus, Larissa Eberle, Victoria Therese Mücke, Julia Bein, Sandra Beyer, Peter J. Wild, Robert Schierwagen, Sabine Klein, Stefan Zeuzem, Christoph Welsch, Jonel Trebicka, Angela Brieger

**Affiliations:** 1Medical Clinic 1, University Hospital Frankfurt, Goethe University, 60590 Frankfurt am Main, Germanyschwarzkopf@med.uni-frankfurt.de (K.M.S.); a.brieger@em.uni-frankfurt.de (A.B.); 2Department of Internal Medicine B, University of Münster, 48149 Münster, Germany; 3Dr. Senckenberg Institute of Pathology, University Hospital Frankfurt, Goethe University, 60590 Frankfurt am Main, Germany

**Keywords:** ACLF, bile duct ligation, carbon tetrachloride, sepsis

## Abstract

Acute-on-chronic liver failure (ACLF) is associated with increased mortality. Specific therapy options are limited. Hypoxia-inducible factor 1 alpha (HIF-1α) has been linked to the pathogenesis of chronic liver disease (CLD), but the role of HIF-1α in ACLF is poorly understood. In the current study, different etiologies of CLD and precipitating events triggering ACLF were used in four rodent models. HIF-1α expression and the intracellular pathway of HIF-1α induction were investigated using real-time quantitative PCR. The results were verified by Western blotting and immunohistochemistry for extrahepatic HIF-1α expression using transcriptome analysis. Exploratory immunohistochemical staining was performed to assess HIF-1α in human liver tissue. Intrahepatic HIF-1α expression was significantly increased in all animals with ACLF, regardless of the underlying etiology of CLD or the precipitating event. The induction of HIF-1α was accompanied by the increased mRNA expression of NFkB1 and STAT3 and resulted in a marked elevation of mRNA levels of its downstream genes. Extrahepatic HIF-1α expression was not elevated. In human liver tissue samples, HIF-1α expression was elevated in CLD and ACLF. Increased intrahepatic HIF-1α expression seems to play an important role in the pathogenesis of ACLF, and future studies are pending to investigate the role of therapeutic HIF inhibitors in ACLF.

## 1. Introduction

Acute-on-chronic liver failure (ACLF)—an acute deterioration of liver function in patients with chronic liver disease (CLD)—is characterized by consecutive organ failure(s) and associated with grim short-term survival [1,2,3]. To date, bacterial infections and ongoing or increased alcohol consumption are the leading precipitating events of ACLF in the Western world [4]. Besides liver transplantation in selected patients and viral suppression therapy in ACLF caused by, e.g., hepatitis B virus flares, no specific treatment options exist today. Supportive care (e.g., early administration of broad-spectrum antibiotics and symptomatic treatment of different organ failures) in specialized intensive care units is the mainstay of ACLF treatment [5,6,7].

Hypoxia-inducible factor (HIF) is a heterodimer transcription factor consisting of an unstable alpha (α) unit and a stable beta unit. Usually, HIF activity is downregulated in normoxia via prolyl-hydroxylases (PHDs) and the asparagine hydroxylase factor-inhibiting HIF (FIH) [8]. During hypoxia, hydroxylase activity is reduced, the HIF-1α subunit stabilizes, and the heterodimers of HIF bind to specific loci, the hypoxia-responsive elements, in the nuclei of cells and influence the expression of a wide range of genes involved in angiogenesis and proliferation, cellular metabolism and inflammation [9,10]. So far, several other mechanisms have been revealed to play a crucial role in HIF-1α activation. Growth factors (e.g., IGF 1) using the PI3K/mTOR or MAPK signaling pathway, proinflammatory cytokines (e.g., interleukin (IL)-6, interferon gamma (IFNγ)) and other bacterial substances using the STAT3 and NFκB signaling pathways are known to influence HIF-1α expression [10,11,12,13].

Recent studies underline the pivotal role of HIF in the pathogenesis of CLD and the development of hepatocellular carcinoma [14,15,16,17,18]. It has been linked to the aggravation of hepatic cell damage and inflammation and the inhibition of liver regeneration and is the major stimulus of angiogenesis and fibrogenesis [19,20]. However, the role of HIF in ACLF remains unclear. This study, therefore, aimed to assess the role of intrahepatic and extrahepatic HIF-1α expression in different rodent models of ACLF.

## 2. Results

### 2.1. General Characteristics

Following the induction of CLD (via bile duct ligation (BDL) and carbon tetrachloride (CCl_4_) plus alcohol), a marked increase in hepatic hydroxyproline content was observed in both models (Figure 1A,F), which was numerically higher in the BDL model and correlated well with the histological examination using Sirius red staining (Figure 1E,J). Animals with CLD showed signs of advanced chronic liver disease with portal hypertension—indicated by a significant increase in spleen/total body weight (2.9-fold and 1.8-fold, respectively, Figure 1B,G) and the development of ascites in 40% and 63% of animals, respectively, Figure 1C,H)—as well as icterus (in 80% and 63% of animals, respectively, Figure 1C,H). The induction of ACLF led to a further deterioration of liver function (Figure 1B,C,G,H) and an increased mortality/dropout rate, which was especially high in the BDL-ACLF group (Figure 1D,I).

### 2.2. Hepatic HIF-1α Expression in CLD and ACLF

Hepatic HIF-1α expression was investigated by performing qPCR on liver tissues. Hepatic HIF-1α mRNA expression increased in both CLD models, reaching significance in the toxic model, whereas it was borderline significant (*p* = 0.06) in the BDL model. Expression further increased following the induction of ACLF in both models (Figure 2A,D). Interestingly, the main driver of increased HIF-1α mRNA levels in the BDL model appeared to be a marked upregulation of NFκB1 and also STAT3 (Figure 2B), while in the toxic model, both NFκB1 and STAT3, as well as an increase in the mRNA expression of PIK3CD and PLCG2, seem to contribute to the observed phenomenon. In ACLF, the mRNA levels of NFκB1 and STAT3 further increased in the BDL model, while a significant decrease in EGLN1 mRNA—catalyzing the hydroxylation of HIF-1α, leading to its proteasomal degradation—was associated with increased HIF-1α mRNA expression. Western blotting confirmed consecutively increasing hepatic HIF-1α protein expression from CLD to ACLF (Figure 2G). Similarly, the increasing protein expression of iκB and piκB reflected the activation of the NFκB pathway at the protein level (Figure 2G). The mRNA expression of genes downstream of HIF-1α, namely, erythropoietin (EPO) and nitric oxide synthase 2 (NOS2), significantly increased in CLD and further increased in ACLF (Figure 2H). Immunohistochemistry was performed to localize the origin of HIF-1α. Interestingly, the only relevant source of HIF-1α was observed mainly in hepatocytes, and an increase in nuclear HIF-1α could be seen in ACLF (Figure 3G).

To further validate our results, qPCR was performed on two other CLD/ACLF models. Intrahepatic HIF-1α mRNA expression was markedly increased in rodents with a systemic infection with (ACLF) and without CLD caused by autoimmune hepatitis, while CLD itself did not influence the intrahepatic HIF-1α mRNA level (Figure 3A–F). Along with increased HIF-1α mRNA expression, elevated mRNA levels of NFκB1 and STAT3 could be observed. In the second model, in which alcohol binges were used to cause ACLF while CLD was induced using a toxic approach, again, intrahepatic HIF-1α mRNA expression was significantly increased in ACLF, but not in CLD, again accompanied by increases in NFκB1 and STAT3 mRNA levels. Interestingly, in the toxic model, a reduction in the EGLN1 mRNA level was observed in ACLF induced by alcohol binges, which was—due to the limited number of animals—only borderline significant (*p* = 0.069).

### 2.3. Hepatic HIF-1α Expression in Human Liver Samples

In an exploratory attempt to validate the observed increase in HIF-1α mRNA expression in rodent models, a limited number of formalin-fixed paraffin-embedded liver samples from patients with liver cirrhosis (CLD), acute-on-chronic liver failure (ACLF) and healthy controls were investigated using immunohistochemistry (Figure 3H). Similar to the rodent models, HIF-1α expression was markedly elevated in stained samples from patients with ACLF (Figure 3H, lower panel), while a relevant increase could also be observed in specimens from patients with CLD (Figure 3H, middle panel). In healthy controls, hepatic HIF-1α expression was less pronounced (Figure 3H, upper panel).

### 2.4. Extrahepatic HIF-1α Expression

Extrahepatic HIF-1α mRNA expression was investigated in a transcriptome analysis of the cholestatic CLD and ACLF rat models. Here, a significantly increased amount of HIF-1α mRNA was observed in the skeletal muscle (both CLD and ACLF) and small bowel tissue (increase in CLD, non-significant decrease in ACLF, Figure 4A), while other extrahepatic organs (brain, heart, kidney, lung, spleen) did not show this phenomenon. We therefore further investigated extrahepatic HIF-1α in the small bowel (Figure 4B–D) and skeletal muscle (Figure 4E–G) via qPCR. HIF-1α mRNA expression was lower in both CLD and ACLF in the small bowel, though not significant. An increase in HIF-1α mRNA was observed in ACLF in the skeletal muscle, yet results were heterogeneous and therefore not significant.

## 3. Discussion

This study describes the intra- and extrahepatic HIF-1α expression in different stages of liver disease, with a focus on its role in ACLF, with the aim of unraveling the possible mechanisms of intracellular HIF-1α induction. Interestingly, intrahepatic HIF-1α expression seems to be a general phenomenon in ACLF across different etiologies and underlying liver diseases. It is expressed mainly in hepatocytes, and its induction is accompanied by increases in NFκB1 and STAT3. Extrahepatic HIF-1α expression seems to play a negligible role in CLD and in ACLF.

Recent studies have emphasized the important role of intrahepatic HIF in the pathogenesis of CLD [17,18]. It has been linked to the aggravation of hepatic cell damage and inflammation and the inhibition of liver regeneration and is the major stimulus of angiogenesis and fibrogenesis [19,20]. For example, HIF-1α is known to be associated with fibrosis induced in mice via BDL, and HIF-1α-deficient mice showed alleviated signs of fibrosis progression [18]. Moreover, hypoxia-induced signaling in both liver sinusoidal endothelial cells (LSECs) and hepatic stellate cells (HSCs) is linked to the promotion of angiogenesis and the development of portal hypertension [19,21]. Accordingly, in both our rat models (BDL and toxic), intrahepatic HIF-1α mRNA expression was already increased at the stage of CLD. Similar results were observed in the toxic mouse model of CLD, while in CLD due to autoimmune hepatitis in mice, HIF-1α mRNA levels were comparable to the control group.

In ACLF, we observed increased intrahepatic HIF-1α expression across all different etiologies and underlying liver diseases: regardless of the underlying etiology of CLD (toxic/alcohol, autoimmune or cholestatic) and the precipitating event of ACLF (infections—sterile via LPS injection or unsterile via stool injection—or toxic via alcohol binges), strong intrahepatic HIF-1α mRNA expression was detected in all qPCRs in both the rat and mouse models. These results were confirmed by Western blotting, and the mRNA levels of downstream genes of HIF-1α were markedly increased. Interestingly, intrahepatic HIF-1α induction was associated with an increase in NFκB and/or STAT3 mRNA, while increased PIK3CD and PLCG2 mRNA expression was observed in some, but not all, models. Signs of NFκB pathway activation on protein levels were reflected by Western blots and correlated well with the results observed on the mRNA level. In the toxic models, diminished EGLN1 mRNA levels could be detected in ACLF, indicating the decoupling of a common HIF feedback loop. While we expected HIF-1α to be distributed among different hepatic cell types, histology revealed a predominant localization of HIF-1α in hepatocytes.

It has to be noted that LPS induces NOS2. Because of that, we decided to include an additional downstream gene of HIF, EPO. Importantly, our data on NOS2 upregulation are in line with current knowledge on the development of organ dysfunction in ACLF, showing inflammation-dependent nitric oxide liberation as a key driver of organ impairment and ACLF [22]. Our results provide further evidence and might at least partly explain a possible mechanism for NO-dependent endothelial dysfunction in ACLF.

Intrahepatic HIF-1α mRNA expression was especially high in the BDL-ACLF model. Here, survival among animals was especially low. One could speculate that the increased mRNA level of HIF-1α may be associated with this phenomenon, as HIF is known to be associated with the aggravation of hepatic cell damage and inflammation and the inhibition of liver regeneration [19,20]; however, the experiments in our study were not designed to prove this association. Interestingly, in human alcoholic liver disease, HIF-1α has also been associated with both hepatoprotective effects, as well as reduced dysbiosis, and an increased intestinal barrier [23,24].

As inhibitors of HIF are now commercially available [25,26,27], future studies using either these inhibitors or experiments with HIF-1α-deficient knockout animals are needed to draw such conclusions. Yet, we know from sepsis models that HIF-1α is a critical determinant of the sepsis phenotype, promoting the production of many inflammatory cytokines (e.g., TNFα, IL-1, IL-4, IL-6 and IL12), and HIF-1α deletion in human macrophages reduces LPS-induced mortality and alleviates the clinical presentation of sepsis, including hypotension and hypothermia [12,28]. Bacterial infections or sterile cytokine storms have been described to play a pivotal role in ACLF pathogenesis [22,29]. In line with this observation is a small preliminary animal study that showed that the inhibitor of HIF1α “genistein” significantly attenuated LPS-induced ACLF in rats [30].

In our study, intrahepatic HIF-1α mRNA expression was accompanied by increases in STAT3 and NFκB mRNA levels, both downstream signaling pathways of inflammatory-cytokine-binding receptors (IL-6R, TLR4 and IFNγR). A limitation of our study is that the study itself was not designed to prove that the increased HIF-1α expression was actually induced by the activation of the STAT3 and NFκB pathways. This causality can only be demonstrated via further experiments utilizing pharmacological inhibitors or knockout animal models of the respective pathway. However, we observed the concomitant activation of the NFκB pathway, suggesting an association in that manner. Interestingly, in our ACLF model, changes in HIF-1α mRNA levels in extrahepatic solid organs remained negligible. Similar to the investigated rodent models, increased HIF-1α levels could be noted in an exploratory analysis of a limited number of human CLD and ACLF liver tissue specimens using immunohistochemistry. However, patients with severe ACLF are usually critically ill when admitted to the hospital, and a liver biopsy is seldom performed, as, for clinicians, little additional information can be obtained via histology. Furthermore, a liver biopsy increases the risk of bleeding in patients with ACLF [31]. Thus, it was difficult to obtain data on the liver tissues in this patient collective. Apart from that, due to its unstable condition, circulating HIF-1α expression could not be measured in either rodents or humans. To overcome both limitations, different animal models were investigated that included common underlying etiologies of human CLD and the main precipitating events triggering ACLF known to date.

In conclusion, increased intrahepatic HIF-1α expression seems to play an important role in the pathogenesis of ACLF, though future experiments are needed to unravel the detrimental role in animals’ and patients’ outcomes. The effect of the recently available HIF inhibitors should be investigated in ACLF triggered by a systemic infection in the future.

## 4. Materials and Methods

### 4.1. Animals

For the experiments, male Sprague Dawley rats (Charles River, Sulzfeld, Germany, initial total body weight, TBW of 180–200 g), aged 6–8 weeks, were used. Rats were fed standard rat chow. All animals received water and chow ad libitum. The animals were kept in individually ventilated cages at 22 °C with a 12 h day/night cycle. All animal experiments were performed in accordance with the German Animal Protection Law and the guidelines of the animal care unit at the Zentrale Forschungseinrichtung (ZFE, University Hospital Frankfurt, House 67, Frankfurt, Germany) and approved by the local authorities (Regierungspräsidium Darmstadt; vote: V54-19c20/15-FK/K6248). An overview of the performed animal experiments is shown in Figure 5A.

### 4.2. Induction of Chronic Liver Disease

As liver cirrhosis may be caused by different etiologies in humans, a similar approach was used in our experiment. For the induction of chronic liver disease (CLD), two different animal models were applied in our study. Bile duct ligation (BDL) was performed to simulate cholestatic liver disease in wild-type (WT) rats, as described previously (*n* = 25) [32]. Briefly, following anesthesia with Ketamine/Xylazine (Sigma-Aldrich, St. Louis, MO, USA), median laparotomy was performed, and the bile duct was gently separated from the portal vein and hepatic artery. It was then ligated with 3-0 silk with two surgical knots, proximal and distal, and then dissected in between. Sham-operated rats served as controls (*n* = 10).

A combination of carbon tetrachloride (CCl_4_, Carl Roth GmbH, Karlsruhe, Germany) and alcohol (Sigma-Aldrich, St. Louis, MO, USA) served as a second model for the induction of CLD, simulating a toxic/alcoholic approach. Rats (*n* = 29) received CCl_4_ intraperitoneally (i.p.) twice weekly (2 µL/g 1:1 corn oil/CCl_4_ in the first week, then 1 µL/g). Drinking water contained 4%, 8% and 16% alcohol in weeks 1, 2 and the following weeks, respectively. Rats receiving corn-oil injections and regular drinking water without alcohol served as controls (*n* = 8).

### 4.3. Induction of Acute-on-Chronic Liver Failure

The induction of ACLF was performed using intravenous (i.v.) lipopolysaccharide injections (LPS from E. coli O111:B4, Sigma-Aldrich, St. Louis, MO, USA, 6.25 µg/kg body weight), and the animals were sacrificed 72 h later. LPS injections were performed after the development of CLD with clinical signs of ascites, icterus and/or weight loss. For BDL rats, LPS injections were performed on days 21 and 25. In the toxic model, injections were performed according to the degree of CLD progression after day 30. An overview of the experimental design is depicted in Figure 1B.

### 4.4. Additional Animal Models

To show the robustness of the results not only across different etiologies of CLD and ACLF but also across different animal species, two mouse models with CLD and ACLF were chosen, and available liver tissue was analyzed. Of note, these animal experiments were not performed specifically for this study. Liver tissue was obtained and kindly provided by collaborating research groups in our lab to confirm and investigate the mechanisms of HIF induction (Figure 5B) [33,34].

In the first of these models, CLD was established by CYP2D6-linked adenovirus (10^10^ plaque-forming units of Ad-2D6 in 0.1 mL of PBS i.p., self-manufactured as described previously [33]) via the induction of autoimmune hepatitis in 6–8-week-old C57BL/6J (Charles River, Sulzfeld, Germany). An i.p. injection of cecal slurry to simulate a systemic infection (*n* = 6 with CLD, infection alone and ACLF each) was used to cause ACLF, as described previously [35]. Untreated mice (*n* = 6) served as controls. The cecal slurry was self-manufactured: the laparotomy of a naïve donor C57BL/6J mouse was followed by the extraction of cecal stool. The slurry was diluted and filtered several times, as described elsewhere [36]. The cecal slurry was injected at a dose of 1 mg cecal slurry/g of TBW (50 mg slurry/1 mL sodium chloride solution).

The second model used C57Bl/6J mice, aged 6–8 weeks (Charles River, Sulzfeld, Germany), and CCl_4_ i.p. injections, as described above (Section 4.2), administered twice weekly induced CLD. Here, *n* = 5 mice received drinking water with 4%, 8% and 16% alcohol in weeks 1, 2 and the following weeks (=CLD), respectively, and *n* = 5 mice did not receive alcohol (CLD*) [34]. To induce ACLF, animals received an additional two binges of alcohol (gavage with 31.5% Vol alcohol, 5 g/kg TBW, ~400 µL for a 20 g mouse with 0.25 g/mL ethanol solution) [37] starting on day 52, with an interval of 3 days in between binges. Mice receiving corn-oil injections and regular drinking water without alcohol served as controls (*n* = 5).

### 4.5. Human Liver Tissue

In an exploratory attempt to correlate the observed rodent HIF-1α protein levels, formalin-fixed and stored human liver tissue samples (*n* = 6, patients with CLD, ACLF and healthy controls who had received a transcutaneous ultrasound liver biopsy during hospitalization in the University Hospital Frankfurt, Frankfurt am Main, Germany) were analyzed via immunohistochemistry to assess hepatic HIF-1α expression in humans in these conditions.

The use of human material was approved by the local ethics committee of the Faculty of Medicine of the University Hospital Frankfurt (Voting-No 4/09, approval year 2009), and all patients and healthy donors gave their written informed consent before inclusion in the study. The study was conducted in accordance with the Declaration of Helsinki.

### 4.6. Animal Tissue and Blood Collection

At the end of the experiments, animals were anesthetized with Ketamine/Xylazine (Sigma-Aldrich, St. Louis, MO, USA, 100 mg/10 mg/kg of TBW), and blood and tissue collection was performed following median laparotomy. Blood samples (in EDTA) and serum samples (AppliChem, Darmstadt, Germany) were obtained from the inferior vena cava. Liver, spleen, muscle tissue (Musculus iliopsoas) and the small intestine were weighed and stored at −80 °C (freezer: Gram Scientific, Vojens, Denmark) until further used, as described previously [34].

### 4.7. Histology

For the detection of collagen fibers, liver specimens were fixed in 4% formalin, paraffin-embedded, and stained in 0.1% Sirius red in saturated picric acid (Chroma, Münster, Germany), as described previously [38].

For immunohistochemical staining to visualize HIF-1α expression, slides were deparaffinized and rehydrated. Following the demasking of antigens, slides were blocked with 10% goat serum (Abbcam, Cambridge, UK) with 1% BSA in TBS (Sigma-Aldrich, St. Louis, MO, USA) for 2 h and incubated overnight with the primary antibody (HIF1α Antibody, 1:200, Novus Biologicals, Minneapolis, MN, USA). A secondary streptavidin-HRP-conjugated antibody was subsequently applied (undiluted, 60 min, R&D Systems, Minneapolis, MN, USA). Finally, slides were developed in diaminobenzidine (Cell Signaling Technology, Cambridge, UK) and counterstained with hematoxylin.

### 4.8. Hepatic Hydroxyproline Content

To evaluate the degree of fibrotic remodeling of the liver, hepatic hydroxyproline content was analyzed in analog liver specimens. Liver tissue (50 mg) was dissolved in 6 M hydrochloric acid (Sigma-Aldrich, St. Louis, MO, USA) at 110 °C and homogenized in TissueLyser LT (Qiagen, Hilden, Germany). Samples were dissolved in methanol (Sigma-Aldrich, St. Louis, MO, USA, 1:1), oxidized with chloramine T 0.84% (Sigma-Aldrich, St. Louis, MO, USA) and finally incubated with Ehrlich’s reagent (1.66 mM, Sigma-Aldrich, St. Louis, MO, USA). The absorption of 150 µL of each specimen was measured photometrically at 558 nm with Multiskan Sky (Thermo Fisher Scientific, Darmstadt, Germany) and compared with the hydroxyproline standard function.

### 4.9. Quantitative Polymerase Chain Reaction

To assess HIF-1α expression levels and explore possible mechanisms of HIF-1α induction, RNA isolation from different organ tissues and consecutive polymerase chain reaction were performed. Organ tissue was homogenized, and RNA was extracted using the TRIzol reagent from Ambion RNA Isolation Kit (Thermo Fisher Scientific, Darmstadt, Germany) according to the manufacturer’s protocol. RNA concentration was measured with a Nano Drop 2000 (Thermo Fisher Scientific, Darmstadt, Germany). cDNA synthesis was performed with the ImProm-II^TM^ Reverse Transcription System (Promega, Madision, WI, USA). DNase digestion was performed to eliminate genomic DNA. Real-time quantitative PCR (qPCR) was performed using TaqMan gene expression assays (Thermo Fisher Scientific, Darmstadt, Germany, Table 1) in accordance with the manufacturer’s protocol: the qPCR reactants included 5 µL of TaqMan Gene Expression Mastermix, 0.5 µL of TaqMan assay and 1 µL of cDNA added with 3.5 µL of RNase-free water (Promega, Madison, WI, USA) in a total volume of 10 µL. A StepOnePlus Real-Time PCR-System (Applied Biosystems, Foster City, CA, USA) was used, and StepOne version 2.0 software was used to measure the qPCR curves. All experiments were performed at least three times. Gene expression was exported to Microsoft Excel and calculated using the 2^−∆∆Ct^ method [39], and the results were standardized against ribosomal RNA 18S as the housekeeping gene. Gene expression levels are depicted as x-fold expression compared to the respective control group.

### 4.10. Western Blotting

Protein levels were analyzed by Western blotting. Briefly, snap-frozen liver samples were homogenized and diluted. The protein content of the homogenates was assessed with the DC assay kit (Bio-Rad, Munich, Germany). For the detection of HiF-1α, 30 ng of recombinant Human HIF-1 alpha protein (#ab154478, Abcam, Cambridge, MA, USA) (positive control), 50 µg of HeLa whole-cell extract (negative control, Abcam, Cambridge, MA, USA) and 100 µg of whole-protein extract from liver tissues were analyzed. For the detection of IκB, as well as pIκB, 30 ng of recombinant Human HIF-1 alpha protein, 50 µg of HEK293T whole-cell extract (as endogenous expression of IκB has been described for HEK293T cells, Santa Cruz, Dallas, TX, USA) [40] and 50 µg of whole-protein extract from liver tissues were used. SDS-PAGE was performed under reducing conditions (10% gels), and proteins were blotted using a wet-blot approach onto nitrocellulose membranes (Thermo Fisher Scientific, Darmstadt, Germany). Next, membranes were blocked and incubated with the primary antibodies against HIF-1α (1:500, NB100-134, 1:500, Novus biologicals, Minneapolis, MN, USA), pIκBα (H.709.9, 1:1000, Invitrogen, Carlsbad, CA, USA), IκBα (T.937.7, 1:1000, Invitrogen, Carlsbad, CA, USA) and β-Aktin (clone AC-15, 1:5000, Sigma-Aldrich, St. Louis, MO, USA), followed by fluorescently labeled secondary antibodies (anti-mouse IRDye^®^ 680 LT or anti-rabbit IRDye^®^ 680 LT, all from LI-COR Biosciences; Lincoln, NE, USA) or unlabeled corresponding secondary antibodies. Finally, blots were developed using fluorescence or enhanced chemiluminescence.

### 4.11. Transcriptome Analysis

From an earlier BDL/ACLF experiment (with the same experimental setup), tissue samples from the spleen, muscle, kidney, duodenum, brain, heart and lung were available to perform transcriptome analysis. Transcriptome analysis was performed by OakLabs (Henningsdorf, Germany) using the Agilent Microarray XS (Agilent Technologies, Santa Clara, CA, USA). Briefly, the Low Input QuickAmp Labeling Kit (Agilent Technologies, Santa Clara, CA, USA) was used to obtain fluorescent complementary RNA (cRNA). Next, hybridization to microarrays with the Gene Expression Hybridization Kit (Agilent Technologies, Santa Clara, CA, USA) was performed, and fluorescent signals were detected by SureScan Microarray Scanner (Agilent Technologies, Santa Clara, CA, USA).

### 4.12. Statistical Analysis

For statistical analysis, BiAS, Version 11.03 (Epsilon-Verlag, Darmstadt, Germany), and GraphPad Prism for Windows (v5.02; Graph Pad Software Inc., San Diego, CA, USA) were used. Data were expressed as the mean ± standard error of the mean (SEM). Groups were tested using the Shapiro–Wilk test for a normal distribution. Comparisons between two groups were performed using the unpaired *t*-test or Mann–Whitney U test.

For the mortality outcome, time-to-event was estimated with Kaplan–Meier method and differences were compared with the Logrank test.

## Figures and Tables

**Figure 1 ijms-25-01542-f001:**
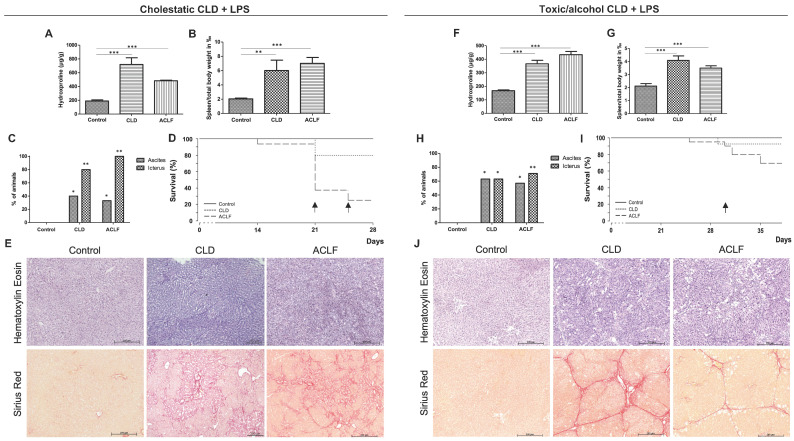
Characteristics of the two animal models. Hydroxyproline content in control rats (*n* = 10), rats with chronic liver disease (CLD, *n* = 5) and rats with acute-on-chronic liver failure (ACLF, *n* = 4) (**A**), spleen-to-total-body-weight ratio as a surrogate parameter for portal hypertension (**B**), rates of ascites and icterus (**C**), Kaplan–Meier curve of survival (**D**) and histology of liver specimens with hematoxylin–eosin and Sirius red staining (**E**) for the cholestatic model and the respective characteristics of the toxic animal model (*n* = 8 controls, *n* = 8 with CLD and *n* = 14 with ACLF) (**F**–**J**). Arrows mark the time of LPS injection (**D**) or the beginning of LPS injections in animals showing clinical signs of progressing CLD (**I**). * *p* < 0.05, ** *p* < 0.01, *** *p* < 0.001.

**Figure 2 ijms-25-01542-f002:**
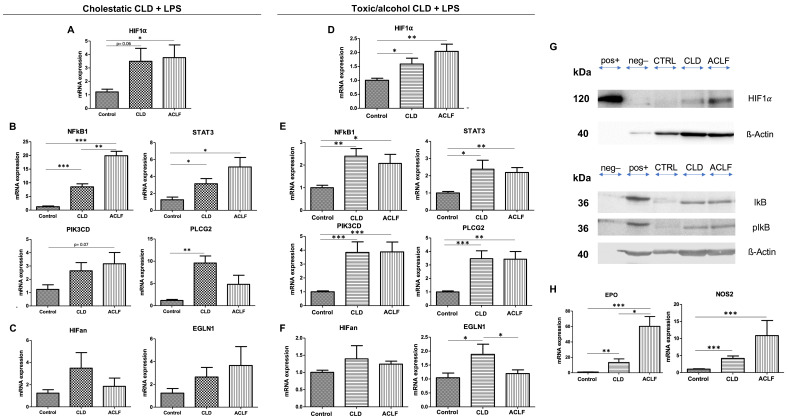
HIF-1α expression in liver tissue. mRNA expression levels of HIF-1α (**A**), inducers of HIF-1α (**B**) and inhibitors (**C**) in the cholestatic model (BDL) causing chronic liver disease (CLD, *n* = 5); acute-on-chronic liver failure was induced by intravenous lipopolysaccharide injection (*n* = 4). Respective mRNA expression levels in the CCl_4_ model causing CLD (*n* = 8 with CLD, *n* = 14 with ACLF) (**D**–**F**). Increased HIF-1α protein expression was confirmed by Western blotting, and signs of NFκB1 pathway activation were reflected by increased expression of IκB and pIκB (**G**). Effects on mRNA expression levels of downstream genes of HIF are depicted in (**H**). Note: (**G**) For the detection of HIF-1α expression via Western blotting, 30 ng of recombinant Human HIF-1 alpha protein (#ab154478, Abcam, Cambridge, MA, USA) as a positive control (pos+), 50 µg of HeLa whole-cell extract as a negative control (neg−) and 100 µg of whole-protein extract from mouse liver tissues (CTRL, CLD and ACLF, respectively) were analyzed. For IκB and pIκB detection, 50 µg of HEK293T cell extract was used (positive control for IκB and pIκB) instead of HeLa extract, and only 50 µg of whole-protein extract from mouse liver tissues was used (CTRL, CLD, ACLF). * *p* < 0.05, ** *p* < 0.01, *** *p* < 0.001.

**Figure 3 ijms-25-01542-f003:**
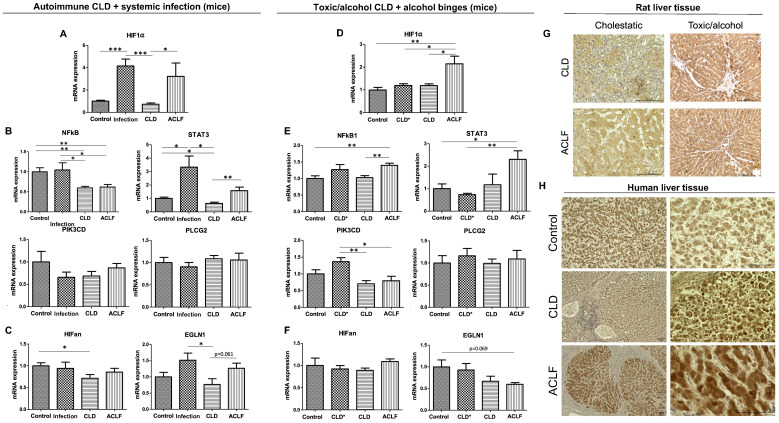
HIF-1α expression in liver tissue of the additional ACLF models. mRNA expression levels of HIF-1α (**A**), inducers of HIF-1α (**B**) and inhibitors (**C**) in the autoimmune hepatitis model causing chronic liver disease (CLD); acute-on-chronic liver failure was induced by systemic infection via intraperitoneal stool injection (*n* = 6 for each group). Respective mRNA expression levels in the CCl_4_ model with (CLD, *n* = 5) and without (CLD*, *n* = 5) additional alcohol (**D**–**F**); ACLF was induced by alcohol binges (*n* = 4). Immunohistochemistry from rat liver tissue showed that HIF-1α was mainly localized in hepatocytes, and increased nuclear HIF-1α could be detected in ACLF (**G**). Immunohistochemistry of human tissue samples from patients with CLD, ACLF and healthy controls showed markedly elevated hepatic HIF-1α expression in ACLF specimens and areas of increased HIF-1α expression in CLD (darker brownish staining), while expression was less pronounced in healthy controls (**H**). * *p* < 0.05, ** *p* < 0.01, *** *p* < 0.001.

**Figure 4 ijms-25-01542-f004:**
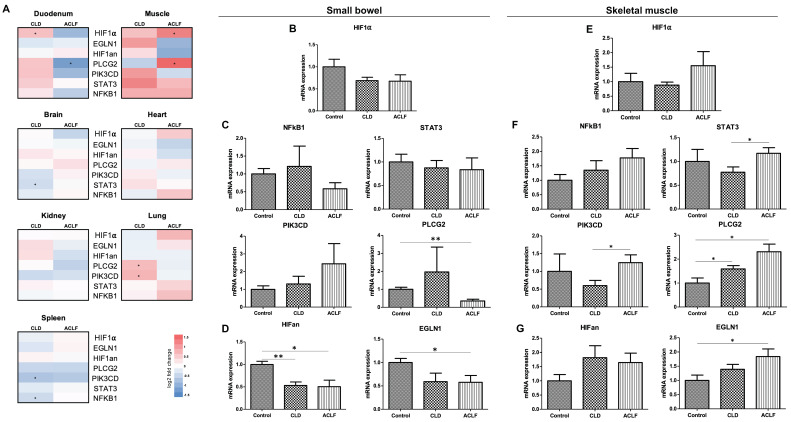
HIF-1α mRNA expression in extrahepatic organs of the BDL model. Heat maps of HIF-1α and inducers and inhibitors of HIF-1α in transcriptome analysis of extrahepatic organs (control vs. CLD and CLD vs. ACLF) (**A**), mRNA expression levels of HIF-1α (**B**), inducers of HIF-1a (**C**) and inhibitors (**D**) in specimens from small bowel. Respective mRNA expression levels in specimens from skeletal muscle tissue (**E**–**G**). * *p* < 0.05, ** *p* < 0.01.

**Figure 5 ijms-25-01542-f005:**
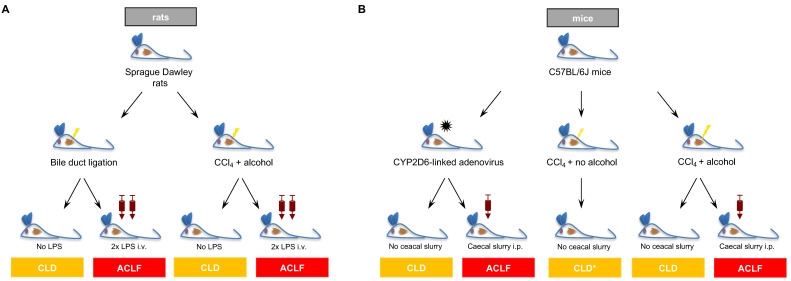
Overview of the experimental design. The original experiments were performed with rats (**A**). Here, chronic liver disease (CLD) was induced via bile duct ligation (BDL) or administration of carbon tetrachloride (CCl_4_) intraperitoneally and alcohol orally. Injection of lipopolysaccharide (LPS) triggered acute-on-chronic liver (ACLF). (**B**) illustrates the additional mouse models from which liver tissues were kindly provided to validate our results in rats.

**Table 1 ijms-25-01542-t001:** TaqMan assays.

Gene	Assay ID	Species
*HIF1A*	Rn01472831_m1	rat
*HIF1AN*	Rn01766292_m1	rat
*ELGN1*	Rn00710295_m1	rat
*STAT3*	Rn00680715_m1	rat
*NFκB1*	Rn01399572_m1	rat
*PLCG2*	Rn01431998_m1	rat
*PIK3CD*	Rn01516709_m1	rat
*NOS2*	Rn00561646_m1	rat
*EPO*	Rn01481376_m1	rat
*18sRNA*	REF: 4308329	rat

## Data Availability

Data are contained within the article.

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
