# Peer review of "The Role of Hypoxia-Inducible Factor 1 Alpha in Acute-on-Chronic Liver Failure"

_ijms, 2024, doi:10.3390/ijms25031542_

Round 1

Reviewer 1 Report

Comments and Suggestions for Authors

The manuscript written by Mucke et al describes the role of HIF1a in Acute-on-chronic liver failure. The study is only descriptive and suffers from shortcomings.

Fig 1a: Should be removed because of poor quality and redundancy. Citation is missing. This figure looks like a screenshot from KEGG.

Fig 1b: I suggest to re-design this figure for clarity. It should be described in more detail which model belongs to which disease condition (CLD, ACLF).

Fig 2: n=? Must be defined in the figure legend.

Fig. 3: n=?

Fig. 3g: The western blot looks weird. Why is there no actin band in the positive control?

The activation of NFkB pathway must be shown by western blotting (eg Ikb degradation) or immunohistochemistry (eg RelA nuclear translocation). Increased expression of NFkB subunits does not mean increased activity of this pathway.

Line 112: The data does not support the conclusion that HIF1a expression is driven by STAT3 and NFkB. This is just an assumption. Regulation of these pathways can be coincidence. There is no causal relationship shown (no NFkB or STAT3 inhibitor).

Comments on the Quality of English Language

English is ok in general.

Author Response

Reviewer #1 comments

The manuscript written by Mucke et al describes the role of HIF1a in Acute-on-chronic liver failure. The study is only descriptive and suffers from shortcomings.

Comment 1: Fig 1a: Should be removed because of poor quality and redundancy. Citation is missing. This figure looks like a screenshot from KEGG.

Answer: We thank the reviewer for his remark. Indeed, we felt that figure would further improve the understanding of the different mechanisms of HIF-1α activation. The figure itself was drawn by the authors but adopted and inspired by KEGG PATHWAY Database and the different reviews on HIF cited in the article. We regret that you feel it to be redundant as we believe that especially for someone not an expert in HIF it gives a nice overview. However, since this figure is not really essential, we deleted figure 1a as requested and referenced the pathway database for more information (p. 2 line 66). Consecutively, the numeration of the figures has changed: 1 = cut, 2=1, 3=2, 4=3, 5=4 and an additional Figure 5 for the methods part was implemented.

Comment 2: Fig 1b: I suggest to re-design this figure for clarity. It should be described in more detail which model belongs to which disease condition (CLD, ACLF).

Answer: We agree with the reviewer. For clarity we redesigned the new Figure 5 (formerly 1B) now illustrating both rodent animal experiments.

We also added the corresponding disease condition in the figure legend:

“The original experiments were performed with rats (A). Here chronic liver disease (CLD) was induced via bile duct ligation or administration of CCl4 (i.p.) and alcohol. Injection of Lipopolysaccharide (LPS) triggered acute-on-chronic liver (ACLF). (B) illustrates the additional mouse models from which liver tissue was kindly provided to validate our results in rats.”

Comment 3: Fig 2: n=? Must be defined in the figure legend.

Answer: We thank the reviewer for this important remark. “n=” were added in Figures legends 1 (formerly Fig. 2) and Fig. 2 (formerly Fig. 3).

Comment 4: Fig. 3: n=?

Answer: We thank the reviewer for this important remark. “n=” were added in Figures legends 1 (formerly Fig. 2) and Fig. 2 (formerly Fig. 3).

Comment 5: Fig. 3g: The western blot looks weird. Why is there no actin band in the positive control?

Answer: We are very sorry, that we didn’t mention some important information. HIF-1α is quite fragile and western blots for HIF-1α are quite difficult (see current literature). Therefore, we used 30ng recombinant Human HIF-1 alpha protein (ab154478) from Abcam as positive control and 50µg HeLa whole cell extract as negative control.  To investigate HIF-1α in liver tissues, we used 100µg of whole protein extract per lane, respectively. Therefore, the western blot shows no β-Actin band in the lane of the positive control, a thin β-Actin band in the negative control and stronger β-Actin bands in lanes where we analyzed liver tissue. We added this information in the Material and Method section and also as a remark in the figure legend (p.4 ll. 231-239).

Comment 6: The activation of NFkB pathway must be shown by western blotting (eg Ikb degradation) or immunohistochemistry (eg RelA nuclear translocation). Increased expression of NFkB subunits does not mean increased activity of this pathway.

Answer: We thank the author for this important remark. First, we change the manuscript to be more specific. We added throughout the manuscript “mRNA” wherever applicable to emphasize, that we are reffering to mRNA levels. Additionally, a new Western blot was performed to show iκB degradation (Figure 2G and p. 3 ll. 165-166). Furthermore, we added a part in our discussion pointing out that we found signs of NFκB pathway activation but still further confirming experiments are needed to proof a causal relationship between HIF activation and the observed phenomena (p. 7, ll. 500-506).    

Comment 7: Line 112: The data does not support the conclusion that HIF1a expression is driven by STAT3 and NFkB. This is just an assumption. Regulation of these pathways can be coincidence. There is no causal relationship shown (no NFkB or STAT3 inhibitor).

Answer: This comment is absolutely correct, and we apologize if the manuscript suggested that we could proof increased HIF expression via STAT3 and NFkB. Of course, only additional experiments (e.g. with knock out mice or an inhibitor) could further proof this assumption. (See also answer to comment 6). We rephrased all concerning statements throughout the manuscript (for example p. 3 l. 163, , p. 4 ll. 243-247, p. 6 ll. 369-373), the abstract  and added this limitation to our discussion (p. 7 ll. 500-506).

Reviewer 2 Report

Comments and Suggestions for Authors

The authors conducted a rather interesting study, which they described and presented in a largely careless manner. I would like to allow myself a number of questions and critical comments.

Introduction, Fig.1 Why not split A and B into two Figures.

B – the scheme only for rats. Why not present 2 diagrams in a separate figure: for rats and for mice?

The 4.   Materials and Methods section is presented very carelessly. In most cases, the manufacturer of the equipment and reagents is not indicated. I propose to supplement it clearly and correctly. This will greatly improve the overall perception of the manuscript.

4.1. Animals. Where obtained from (supplier), age. What is the reason for such an unequal number of animals in groups? Rats: 25/10, 29/8, mice 5/6 (?).

4.4 – very sloppy. What kind of mice, where are they from, how were they kept? The number in groups is 5 and 6. Why? Is this enough for statistics?

CYP2D6-linked adenovirus – manufacturer?

Cecal slurry - whose, how was it prepared?

alcohol gavage with 31,5% Vol - did the mice actually drink this? The entire volume? Or were these intragastric injections?

4.5 What kind of patients? Was there informed consent? How were liver tissue samples obtained?

4.6 Ketamine/Xylazine – manufacturer? Test tubes, scales, freezer – manufacturer?

2.           Results

In the survival curves, it is logical to indicate with arrows the moment of LPS administration.

What determines the timing of LPS administration and the period (72 hours) after which the animals were euthanized? Why don’t you take into account that by this time (after intravenous administration of LPS at this dose) NOS2 (inducible) has a huge impact on the animal’s vessels, condition and mortality?

Fig.4 Well, at least write that these are mice on the left.

2.3 and Fig. 4G Where did healthy control come from?

2.4 Please clarify, are these rats, mice or patients?))

Discussion

Perhaps the phenomenon was not confirmed in the mouse model due to your negligence. 

Comments on the Quality of English Language

There are flaws in the text, which are probably associated with some general carelessness in the presentation of the material. Careful reading will allow you to easily eliminate these shortcomings. 

Author Response

Reviewer #2 comments

The authors conducted a rather interesting study, which they described and presented in a largely careless manner. I would like to allow myself a number of questions and critical comments.

Comment 1: Introduction, Fig.1 Why not split A and B into two Figures.

Answer: We thank the reviewer for this good idea. Reviewer 1 thought that Figure 1A should be removed because the information provided would be redundant. We therefore removed 1A and added an additional experimental overview (mouse experiments) as part B (new Figure 5A and B, formerly Figure 1B).

Comment 2: B – the scheme only for rats. Why not present 2 diagrams in a separate figure: for rats and for mice?

Answer: Please see answer to your comment 1. For clarification, we added in the revised version of this manuscript an overview of the mouse experiments in the new Figure 5B.

Comment 3: The 4. Materials and Methods section is presented very carelessly. In most cases, the manufacturer of the equipment and reagents is not indicated. I propose to supplement it clearly and correctly. This will greatly improve the overall perception of the manuscript.

Answer: We thank the reviewer for this remark. Indeed, the material and methods sections needed an additional work-up. We updated the manufacturer of the equipment and reagents in all cases where they missing before (see throughout the “4. Materials and Methods section”).

Comment 4: 4.1. Animals. Where obtained from (supplier), age. What is the reason for such an unequal number of animals in groups? Rats: 25/10, 29/8, mice 5/6 (?).

Answer: We added the following detailed information on supplier and age in the manuscript (throughout the method section). Regarding the “unequal number”: First, rat experiments were performed, as the extend of HIF expression was not known and an initial establishment of the model was taken into account, we calculated with higher numbers. Indeed, when you take a closer look at the mortality rate of the LPS-ACLF model (survival ~ 20-30%) material for WB analysis was available in only 4 ACLF and 5 CLD rats. As each group needed 2-3 control animals the number of control animals remains high. Similarly, the CCl4 model was calculated but here mortality rate was much lower and the model could be established more easily that more animals were available. Of note, the mice experiments were not specifically performed for this paper. Liver tissue from these models were kindly provided from another research group of our lab (as this seems not to be clear, we clarified this in the revised version of our paper (p. 8 ll. 584-587). As we already knew statistics would show a significant increase in HIF in the rat models we requested only liver tissue from one set of experiments which were 5/6 mice.”

Comment 5: 4.4 – very sloppy. What kind of mice, where are they from, how were they kept? The number in groups is 5 and 6. Why? Is this enough for statistics?

Answer: We agree with the reviewer and apologize for not providing more detailed information. The initial reason for only referring to the other publication is, that (also see answer to comment #4). these mouse experiments were not specifically performed for this paper. Liver tissue from these models were kindly provided from another research group of our lab (as this seems not to be clear, we clarified this in the revised version of our paper (p. 8 ll. 583-586). We thought that it should be therefore only referenced to the work of these groups. However, we provided additional information to the reader in a revised version (p. 8, ll. 587-640). As we already knew statistics would show a significant increase in HIF in the rat models, we requested only liver tissue from one set of experiments which were 5/6 mice.

Comment 6: CYP2D6-linked adenovirus – manufacturer?

Answer: We thank the reviewer for this question, we revised the Materials and Methods section to provide these information (p. 8, ll. 588-597).

Comment 7: Cecal slurry - whose, how was it prepared?

Answer: We revised the Materials and Methods section to provide these information (p. 8, ll. 592-596).

Briefly, cecal slurry was self-manufactured: Laparotomy of a naïve donor C57BL/6J mouse was performed to extract cecal stool. Then the slurry was diluted and prepared through different filtration processes as described elsewhere.1

Comment 8: alcohol gavage with 31,5% Vol - did the mice actually drink this? The entire volume? Or were these intragastric injections?

Answer: We provided additional information in our revised version of the manuscript (p. 9, ll. 634-639). The binge was performed as a gavage with 31.5% Vol alcohol, 5g/kg TBW, ~400µl for a 20g mouse with 0,25g/ml ethanol solution. A picture and detailed description can be found at Bertola et al., “Mouse model of chronic and binge ethanol feeding (the NIAAA model)”, Nat. Protocols (PMID: 23449255).2

Comment 9: 4.5 What kind of patients? Was there informed consent? How were liver tissue samples obtained?

Answer: We thank the reviewer for this question. Patients received a transcutaneous ultrasound liver biopsy during hospitalization in the University Hospital Frankfurt, Frankfurt am Main Germany. Samples were part of a prospective study which was conducted from 2009 at our hospital and which was approved by the local ethics committee of the Faculty of Medicine of the University Hospital Frankfurt (Voting-No 4/09, approval year 2009) and all patients and healthy donors gave their written informed consent before inclusion in the study. The study was conducted in accordance with the Declaration of Helsinki.

Comment 10: 4.6 Ketamine/Xylazine – manufacturer? Test tubes, scales, freezer – manufacturer?

Answer: We thank the reviewer for this remark. Indeed, the Material and Methods sections needed an additional work-up. We updated the manufacturer of the equipment and reagents in all cases were they missing before (see throughout the Material and Methods section).

Comment 11: In the survival curves, it is logical to indicate with arrows the moment of LPS administration.

Answer: We added arrows to indicate the moment of LPS administration (Figure 1D and I).

Comment 12: What determines the timing of LPS administration and the period (72 hours) after which the animals were euthanized? Why don’t you take into account that by this time (after intravenous administration of LPS at this dose) NOS2 (inducible) has a huge impact on the animal’s vessels, condition and mortality?

Answer: We thank the reviewer for this very important methodical question. The generation of ACLF is difficult and the animal model for the induction of ACLF with LPS was previously established and has been already published and described in detail. The authors of the present study neither changed the timing or periods of LPS treatment but used the same conditions. The ACLF animal models were designed to mimic human disease conditions, particularly systemic inflammation and organ dysfunction in ACLF. Thus, the first LPS dose (day 21 in BDL) was used to induce acute decompensation, while the second dose (day 25 / 72h hours before sacrifice) was used for ACLF induction. A period of 72h after LPS was chosen to allow the development of organ impairment and a full-blown experimental ACLF at the time of sacrifice. As previously shown, these animal models display specific inflammatory patterns that correlate with human liver cirrhosis at different time-points: (1) acute decompensation / recompensation in BDL + LPS on day 21 and (2) ACLF in BDL + LPS on day 21 + 25 and therefore were considered suitable for the analyses.3-5

It is absolutely correct that LPS induces NOS2 and we are totally aware that this also impacts animals’ vessels, condition and mortality. Since our data were verified in different models, we initially didn’t discuss potential side effects of LPS in this manuscript. Importantly, our data on NOS2 upregulation are in line with current knowledge on the development of organ dysfunction in ACLF showing inflammation-dependent nitric oxide liberation as a key driver of organ impairment and ACLF.6 Our results deliver further evidence and might at least partly explain a possible mechanism for NO-dependent endothelial dysfunction in ACLF. We added this information in our discussion section (p. 6 ll. 401-406)

Comment 13: Fig.4 Well, at least write that these are mice on the left.

Answer: We added the information in the figure (now Figure 3, formerly Figure 4).

Comment 14: 2.3 and Fig. 4G Where did healthy control come from?

Answer: We apologize that this important information was not included in the manuscript. In the revised version we added:

“Patients received a transcutaneous ultrasound liver biopsy during hospitalization in the University Hospital Frankfurt, Frankfurt am Main Germany Comment. Samples were part of a prospective study (including healthy controls) which was conducted from 2009 at our hospital and which was approved by the local ethics committee of the Faculty of Medicine of the University Hospital Frankfurt (Voting-No 4/09, approval year 2009) and all patients and healthy donors gave their written informed consent before inclusion in the study. The study was conducted in accordance with the Declaration of Helsinki.“

Comment 15: 2.4 Please clarify, are these rats, mice or patients?

Answer:  When referring to the cholestatic CLD and ACLF model these animals are rats. We added this information to this paragraph for better understanding (p. 6, l. 382).

Comment 16: Perhaps the phenomenon was not confirmed in the mouse model due to your negligence. 

Answer: Unfortunately, we are unsure what the reviewer is referring to in this comment. All experiments were conducted with great vigilance and according to international standards. The mouse models both confirmed an increased HIF-1α expression and, similarly to the rat models, showed enhanced STAT3 and NFkB mRNA level. In case you were referring to the statement, that we did not observe increased HIF-1α expression at the stage of CLD in the autoimmune model (but in ACLF), we rephrased this section of the discussion to make the message clearer to the reader (p. 6, ll. 383-385).  

REFERENCES

  1. Brook B, Amenyogbe N, Ben-Othman R, et al. A Controlled Mouse Model for Neonatal Polymicrobial Sepsis. J Vis Exp 2019(143).
  2. Bertola A, Mathews S, Ki SH, Wang H, Gao B. Mouse model of chronic and binge ethanol feeding (the NIAAA model). Nat Protoc 2013;8(3):627-37.
  3. Praktiknjo M, Schierwagen R, Monteiro S, et al. Hepatic inflammasome activation as origin of Interleukin-1alpha and Interleukin-1beta in liver cirrhosis. Gut 2021;70(9):1799-1800.
  4. Queck A, Bode H, Uschner FE, et al. Systemic MCP-1 Levels Derive Mainly From Injured Liver and Are Associated With Complications in Cirrhosis. Front Immunol 2020;11:354.
  5. Schierwagen R, Uschner FE, Ortiz C, et al. The Role of Macrophage-Inducible C-Type Lectin in Different Stages of Chronic Liver Disease. Front Immunol 2020;11:1352.
  6. Arroyo V, Angeli P, Moreau R, et al. The systemic inflammation hypothesis: Towards a new paradigm of acute decompensation and multiorgan failure in cirrhosis. J Hepatol 2021;74(3):670-685.

Round 2

Reviewer 1 Report

Comments and Suggestions for Authors

The authors addressed all concerns in the revised manuscript.

Comments on the Quality of English Language

ok

Reviewer 2 Report

Comments and Suggestions for Authors

The authors responded to all comments and took into account the recommendations as much as possible.